# A Practical Guide to All-Atom and Coarse-Grained Molecular Dynamics Simulations Using Amber and Gromacs: A Case Study of Disulfide-Bond Impact on the Intrinsically Disordered Amyloid Beta

**DOI:** 10.3390/ijms25126698

**Published:** 2024-06-18

**Authors:** Pamela Smardz, Midhun Mohan Anila, Paweł Rogowski, Mai Suan Li, Bartosz Różycki, Pawel Krupa

**Affiliations:** Institute of Physics Polish Academy of Sciences, Al. Lotników 32/46, 02-668 Warsaw, Poland; psmardz@ifpan.edu.pl (P.S.); midhun@ifpan.edu.pl (M.M.A.); progowski@ifpan.edu.pl (P.R.); masli@ifpan.edu.pl (M.S.L.); rozycki@ifpan.edu.pl (B.R.)

**Keywords:** molecular modeling, molecular dynamics, all-atom force fields, coarse-grained force fields, Amber, SIRAH, Martini, simulation protocols, amyloid beta, intrinsically disordered proteins

## Abstract

Intrinsically disordered proteins (IDPs) pose challenges to conventional experimental techniques due to their large-scale conformational fluctuations and transient structural elements. This work presents computational methods for studying IDPs at various resolutions using the Amber and Gromacs packages with both all-atom (Amber ff19SB with the OPC water model) and coarse-grained (Martini 3 and SIRAH) approaches. The effectiveness of these methodologies is demonstrated by examining the monomeric form of amyloid-β (Aβ42), an IDP, with and without disulfide bonds at different resolutions. Our results clearly show that the addition of a disulfide bond decreases the β-content of Aβ42; however, it increases the tendency of the monomeric Aβ42 to form fibril-like conformations, explaining the various aggregation rates observed in experiments. Moreover, analysis of the monomeric Aβ42 compactness, secondary structure content, and comparison between calculated and experimental chemical shifts demonstrates that all three methods provide a reasonable choice to study IDPs; however, coarse-grained approaches may lack some atomistic details, such as secondary structure recognition, due to the simplifications used. In general, this study not only explains the role of disulfide bonds in Aβ42 but also provides a step-by-step protocol for setting up, conducting, and analyzing molecular dynamics (MD) simulations, which is adaptable for studying other biomacromolecules, including folded and disordered proteins and peptides.

## 1. Introduction

Proteins are one of the most important biological macromolecules, playing a variety of roles in organisms, such as serving as building blocks, catalyzing biochemical reactions as enzymes, facilitating molecular transport as transporters, and governing essential cellular processes as regulatory molecules [1]. For a considerable time, it was thought that every protein possessed a single unique structure, which was considered crucial for its functional role and believed to be deducible solely from its sequence (Anfinsen Dogma [2]). However, with advancements in science, it was discovered that many proteins in physiological conditions contain disordered regions or exist entirely in a disordered state [3]. Such systems are referred to as intrinsically disordered proteins (IDPs), although it may be more precise to classify them as ‘multi-structure proteins’ with low-energy conformational transitions or simply ‘intrinsically flexible systems’ [4]. This behavior is somewhat similar to the fluxional character of some molecules, which is typically used to describe small molecules that undergo rapid interconversion between structurally distinct but energetically equivalent isomers, a process also known as degenerate rearrangement [5]. However, conformations of IDPs are not energetically equivalent, and transitions between them are more complicated in nature due to the larger size of proteins. Nevertheless, due to the analogy, some works are naming the behavior of some metalo- and disordered proteins as fluxional [6,7,8].

Studying IDPs presents challenges due to their highly dynamic nature and the strong dependence of their properties on various environmental conditions, including pH, temperature, redox potential, and the presence of other molecular entities such as receptors or ligands [9]. They are also prone to oligomerization, aggregation, or nanodroplet formation [10]. One example of such a system is the monomeric form of 1–42 amyloid-β (Aβ42), a fundamental component of toxic Aβ42 oligomers [11] and fibrils associated with neurodegenerative diseases, including Alzheimer’s disease [12]. An interesting approach to expedite the study of Aβ fibril formation is the mutation of residues Leu17 and Leu34, which can be efficiently cross-linked in the fibril form, to cysteines with subsequent disulfide bond formation, forcing the peptide conformation to obtain a mature fibril structure without disturbing the fibril formation process [13]. A similar effect is also observed in other polypeptides involved in neurodegenerative diseases, such as α-synuclein, in which the addition of a disulfide bond to the C-terminus significantly increases the mechanical and thermal stability of the fibrils [14].

Disulfide bonds are the most common post-translational modification of proteins, occurring in over 20% of proteins with known structures [15]. They fulfill a myriad of roles, including structural and enzymatic stabilization, cycle regulation, and response to stress conditions, and are commonly used in industry [16]. Despite their high importance, disulfide bonds are often marginalized or even omitted in protein studies. In this work, methods for simulating the presence or absence of disulfide bonds in proteins are discussed. In addition, a novel approach allowing for dynamic formation and disruption of disulfide bonds during simulations based on finite distance restraints [17] is presented, which is most useful for studying disulfide bond breaking and reforming upon mechanical or environmental stress. It should be noted that this approach is of significantly lower resolution than ab initio MD [18,19], such as Born-Oppenheimer or second-generation Car–Parrinello MD [20] implemented in software packages such as CP2K [21] or Quantum Espresso [22], quantum mechanics/molecular mechanics (QM/MM), which can be run in Amber and Gromacs combined with the interface of CP2K or other tools, such as Quick [23], and reactive force field approaches, such as ReaxFF [24,25,26].

MD simulations have emerged as a powerful tool for investigating the structural, mechanical, and dynamical properties of macromolecules. In particular, detailed information about molecular motions at atomic resolution, as obtained from MD simulations, provides valuable insights into the conformational landscapes and dynamics of IDPs. Among the numerous options and parameters available in molecular dynamics (MD) simulations, three of them hold paramount importance and can significantly impact the results: (i) the resolution of the molecular model, (ii) the force field, and (iii) the sampling method [27]. Together, they influence how efficiently the conformational space of the studied system is explored and whether the obtained results are reliable. While most force fields for proteins have been developed over many years primarily based on well-structured molecules, the study of IDPs necessitates the consideration of methods optimized and well-benchmarked for such systems [28,29]. For instance, it has been demonstrated that older force fields, such as ff99SB [30], ff14SB [31], CHARMM22 [32], and CHARMM36 [33], yield results that more deviate from experimental data than newer parameter sets, such as ff19SB [34] and CHARMM36m [35]. Notably, parameters for elements other than proteins, particularly water [36] and ions [37], are especially significant in the case of studies involving IDPs [38]. To achieve satisfactory sampling of the conformational space, more advanced techniques, such as replica exchange MD [39] or conventional MD simulations consisting of multiple independent trajectories, each of reasonable duration, should be employed [40]. However, running simulations with more and longer trajectories can be very costly, especially for large systems comprising more than a single protein chain. Consequently, one can leverage powerful supercomputers or GPUs [41] to accelerate classical all-atom simulations or opt for simplified models, including coarse-grained ones, in which groups of atoms are represented by single interaction centers or beads. Such simplifications not only expedite the calculation of a single MD step but also permit the use of longer time steps and, by reducing the number of degrees of freedom, smooth out the energy landscape, thereby accelerating processes within the simulation [42]. However, most coarse-grained force fields are based on statistical analyses of databases and are thus less suitable for studying IDPs, which are underrepresented in the training data [43]. One approach to addressing this issue is the utilization of a physics-based coarse-grained force field, such as UNRES [44], a component of the UNICORN package [45], or coarse-grained methods developed for IDPs, such as AWSEM-IDP [46]. Nevertheless, with the development of the Martini model [47], the most popular coarse-grained force field to date [47], it has been discovered that accurate scaling of interactions between solutes and solvents can be a key factor enabling MD studies of IDPs [48]. As a matter of fact, it has been recognized that MD simulations with the Martini force field typically generate IDP conformations that are more compact than those observed in experiments. Importantly, it is possible to diminish the discrepancies between simulations and experiments by carefully scaling the protein-water interactions in the Martini force field.

In this work, we present comprehensive protocols for conducting molecular dynamics simulations of IDPs and their mutants, using the monomeric form of amyloid-β42 (Aβ42) and its L17C/L34C mutant (Aβ42_disul_) as illustrative examples. The protocol covers the application of both all-atom and coarse-grained force fields in two widely used simulation packages: Amber [49] and Gromacs [50]. For the all-atom simulations in Amber, we employ the state-of-the-art ff19SB force field [34] coupled with the OPC water model [51], which has been shown to perform well for both folded and disordered proteins and peptides [52,53,54]. We also demonstrate the use of the coarse-grained SIRAH [55] force field in Amber. In Gromacs, we showcase the application of the popular coarse-grained Martini 3 force field [47], with corrections for protein-water interactions [48] to better capture the behavior of IDPs, as well as the SIRAH force field [55]. Our protocol not only guides users through the setup and execution of these simulations but also introduces a novel approach for treating disulfide bonds dynamically, allowing them to form and break during the course of the simulation [56]. This dynamic treatment is achieved through the use of distance restraints, providing a more realistic representation of the disulfide bond behavior compared to the conventional static approach. To validate our protocol and demonstrate its utility, we present new results comparing the conformational dynamics of Aβ42 and Aβ42_disul_ using the various force fields and simulation packages. Our findings clearly illustrate the impact of the disulfide bond on the structural similarity between the monomeric form and the fibrillar state of Aβ42. Furthermore, we highlight the differences in the results obtained from the different methods employed in this study, providing valuable insights into their strengths and limitations for investigating IDPs.

## 2. Results

All of the results described in this manuscript are based on a specific choice of models and their parameters, all of which are described in the Section 4 and in Appendix B and Appendix C, which should provide reasonable results for IDPs and folded proteins. However, the model parameters may be easily modified by editing input files, e.g., to switch the water model, scale the water-protein interactions, or change the force field. Execution of the example scripts provided in the Appendix A generates a complete workflow for preparing input files for MD simulation, performing minimization, equilibration, and production runs, as well as conducting basic analyses, as illustrated in Figure A1. Appendix D contains the technical checks needed to test the convergence of the simulations and to ensure that the periodic boundary box used in the simulations is of sufficient size to prohibit interactions between periodic images of the solute. These tests, although technical in nature, are necessary before performing standard analyses of the results. Having performed the initial verification analysis steps, as described in Appendix D, a proper analysis of the results can be conducted.

### 2.1. Structural Properties of the Monomeric Aβ42 Captured by the Force Fields

The radius of gyration (R_g_) and the maximum dimension (D_max_) are useful quantities that are independent of a reference structure and can usually be easily compared with other computational and experimental measurements. The presented results show that in the all-atom simulations with the ff19SB force field, Aβ42 tends to adopt a rather compact state with an average R_g_ of 1.29 ± 0.14 nm (Table 1). In the SIRAH coarse-grained model, Aβ42 exhibits an even more compact state, with an average R_g_ of 1.13 ± 0.11 nm. This suggests that the SIRAH force field may favor more compact conformations and underestimate the structural diversity of the peptide compared to the all-atom ff19SB force field. On the other hand, the Martini 3 force field generated the least compact conformations on average, with an average R_g_ of 1.32 ± 0.01 nm, which is not much different from the ff19SB results. This finding is further supported by the average values of D_max_, which are the highest for Amber ff19SB, slightly higher than for Martini 3, and much lower for SIRAH (Table 1). This indicates that Martini 3 allows Aβ42 to explore a wider range of conformations, thus better capturing the intrinsically disordered nature of the monomeric peptide.

The analysis of R_g_ histograms (Figure 1) provides insights into the conformational preferences of monomeric Aβ42 in the different force fields. In the Martini 3 force field, a broad single-peak distribution is observed, indicating that monomeric Aβ42 explores a wide range of conformations without strong preferences for specific structures. This suggests that Martini 3 captures well the intrinsically disordered nature of monomeric Aβ42. In contrast, the R_g_ histograms obtained from the simulations with the Amber ff19SB and SIRAH force fields exhibit multi-peak behavior. This indicates that in these force fields, monomeric Aβ42 has a tendency to form semi-stable conformations, sampling distinct structural states. The presence of multiple peaks suggests that certain conformations are energetically favored and more frequently visited during the simulations. Notably, the main peak of the R_g_ distribution in SIRAH is observed at a very low R_g_ value of around 1.0 nm, and the overall R_g_ distributions span the smallest range among the force fields studied. This finding implies that the SIRAH force field may not fully capture the disordered character of monomeric Aβ42, as it appears to favor more compact conformations and underestimate the structural diversity of the peptide. It should be noted that due to the coarse-grained representation and longer computational time, the Martini simulations yield much more rapid and frequent jumps of both R_g_ and D_max_ than the all-atom simulations (see Appendix D); however, the average values are very similar in these methods, with only SIRAH generating more compact conformations (Table 1).

The distributions of maximum dimension (D_max_) obtained from the simulations (Figure 2) exhibit similar trends to the R_g_ distributions, but are generally more uniform. While the R_g_ distributions obtained from the Amber ff19SB and SIRAH force fields show multiple peaks, indicating semi-stable conformations, the D_max_ distributions are generally smoother. The presence of multiple peaks in the R_g_ distributions suggests energetically favored conformations that are more frequently sampled, while the smoother D_max_ distributions highlight the inherent flexibility of the monomeric Aβ42, with significant portions of the peptide remaining highly dynamic and disordered.

Analysis of the solvent accessible surface area (SASA) shows that the highest SASA values are obtained for the Martini force field, while the lowest values are observed for SIRAH. Interestingly, the Amber ff19SB force field generated the average SASA values much closer to those from SIRAH than from Martini. This finding is noteworthy as it does not fully correlate with the compactness of the monomeric Aβ42 captured by the R_g_ and D_max_ values and demonstrates a higher solubility of the Aβ42 in the Martini 3 than in the other force fields.

Energy landscapes can provide valuable insights into how well each force field captures the intrinsically disordered nature of the monomeric Aβ42. Analysis of the R_g_ versus end-to-end distance maps (Figure 3) shows that the Martini 3 force field exhibits a very shallow and broad energy minimum, corresponding to an ensemble of conformations with many quite extended structures. This suggests that Martini effectively represents the highly flexible and disordered character of the monomeric Aβ42, allowing it to explore a wide range of conformations without strong preferences for specific structures. The Amber ff19SB force field shows similar behavior to Martini, but with a slightly deeper and narrower energy minimum. This indicates the presence of some semi-stable conformations in the simulations, implying that Amber ff19SB captures the disordered nature of Aβ42 to a slightly lesser extent compared to Martini. However, the energy landscape is still relatively broad, suggesting that Amber ff19SB allows for significant conformational flexibility. In contrast, the SIRAH force field displays a rather narrow energy landscape with multiple deep energy minima. This indicates a strong tendency for the monomeric Aβ42 to form specific conformations, particularly those where the N- and C-termini are in close proximity. Thus, SIRAH may not fully capture the intrinsically disordered character of the monomeric Aβ42. However, it should be noted that in Martini there is also a population of conformations, in which there are interactions between the termini; however, they are much less frequent. Overall, the energy landscape analysis provides further evidence that the Martini force field most accurately represents the disordered nature of monomeric Aβ42, followed by Amber ff19SB, while SIRAH appears to have limitations in capturing the full extent of the Aβ42 intrinsic disorder.

Distributions of the secondary structure content per residue show that both the all-atom Amber ff19SB and coarse-grained SIRAH force fields predict the same amino acid residues to have a preference for forming β-strands: residues 16–21 and 30–35 (Figure 4). However, this propensity is higher in the all-atom Amber ff19SB simulations and covers slightly wider ranges, including residues 30–37 and additionally 38–40. Quantitatively, the Amber ff19SB force field predicts β-strand content of approximately 60–80% for residues 16–21 and 30–40, while SIRAH yields around 40–60% β-strand content for residues 16–21 and 30–35. These results indicate that the all-atom Amber ff19SB force field captures a stronger tendency for these regions to adopt β-strand conformations compared to the coarse-grained SIRAH model. The tendency to form α-helices by the monomeric Aβ42 is noticeable only in the all-atom Amber ff19SB simulations, mainly for residues 13–17 and 23–27, with the α-helical propensities reaching around 20–25%. In contrast, the coarse-grained SIRAH force field does not predict any significant α-helical content for the monomeric Aβ42, while in Martini 3 simulations, no secondary structure content is observed.

### 2.2. Impact of Solute-Solvent Interaction Scaling on the Monomeric Aβ42 in Martini 3 Force Field

In this work, we also present a novel approach to scale interactions between solute and solvent in the Martini force field, which significantly impacts the compactness of the Aβ42 (Figure 5). As one would expect, the average R_g_ increases monotonically with increasing the water-protein interaction rescaling parameter, λ, both in the case of Aβ42 and Aβ42_disul_. Indeed, enhancing the protein-water interactions should result in expanded conformations of IDPs [48,57], but this has not been tested so far for the monomeric Aβ42. Importantly, by comparing the average R_g_ from the simulations with the R_g_ from the experiments, one can determine the optimal value of parameter λ that yields the best agreement between the simulation and experiment [48].

Distributions of the end-to-end distance and R_g_ (Figure 6) clearly show that relatively small changes in the solute-solvent interactions, corresponding to a 2% reduction or increase in λ, do not significantly alter the overall behavior of the system. However, even these small variations in λ can result in a shift in the compactness of the structure by approximately 10% (Figure 5 and Figure 6, Table 2). Therefore, applying a small solute-solvent scaling may be safely used to achieve better agreement with experimental data while preserving the overall behavior of the system. This approach allows one to fine-tune the Martini 3 force field to more accurately reproduce experimental observables related to the compactness of such IDPs as the monomeric Aβ42, while still preserving the key characteristics and dynamics of the system.

### 2.3. Impact of the Disulfide Bond Addition to the Monomeric Aβ42

The L17C/L34C mutant of Aβ42 with an engineered disulfide bond (Aβ42_disul_) is an experimental construct that facilitates studies of the initial aggregation steps of Aβ42 without significantly altering the fibril morphology. Our simulations reveal that Aβ42_disul_ adopts more compact structures compared to the wild-type Aβ42 (Table 1) across all force fields tested, with this effect being most pronounced in the Martini force field. Larger differences are observed in D_max_, indicating that while the overall compactness is not significantly altered, the presence of the disulfide bond prevents Aβ42_disul_ from adopting the most extended conformations. Interestingly, despite the tendency of Aβ42_disul_ to form more compact structures, it exhibits slightly higher solubility in the Amber ff19SB and SIRAH force fields, as evidenced by the increased SASA values. This trend is less apparent in the simulations with the Martini 3 force field, where the more substantial decrease in R_g_ has a greater impact on the SASA. However, the decrease in SASA is proportionally smaller than the decrease in R_g_ (7.8% versus 10.0%), suggesting that a similar solubility-enhancing effect is still present.

Only small differences between average R_g_ values with and without the disulfide bond are observed for the Amber ff19SB and SIRAH force fields. This is contrary to Martini, in which the presence of a disulfide bond decreased the average radius of gyration by about 10%. This effect is caused by the propensity of Aβ42 to form β-hairpins by central and C-terminal amino-acid residues, thereby stabilizing the interactions in which the disulfide bond is being added in the first two force fields, which is not observed in Martini (Figure 7 and Table 3). Visualization of the most extended and most compact structures obtained from the all-atom and coarse-grained simulations with and without the disulfide bond clearly shows that Aβ42 can unfold almost completely in Martini when the disulfide bond is not present, while in Amber ff19SB and SIRAH, some stabilizing interactions are always present. Introduction of the disulfide bond prevents the complete unfolding of monomeric Aβ42 in Martini, while it does not significantly impact the conformation of the peptide in Amber ff19SB and SIRAH. What is a visualization of the impact of the disulfide bond on the R_g_ values (Table 1 and Figure 1).

A hierarchical agglomerative clustering was performed to identify the most abundant conformations of the monomeric Aβ42 sampled during the simulations using the different force fields (Figure 8). Then similarities and differences between these most populated conformations (cluster centroids) were assessed by calculating the root-mean-square deviation (RMSD) of atomic positions (Appendix A). The RMSD analysis reveals that each simulation method generated a diverse set of structures. Some of these structures are found to be relatively similar to those obtained by the other methods, with pairwise RMSD values ranging from 4.7 to 21.1 Å. However, the conformations sampled in the SIRAH simulations exhibit the greatest differences compared to those obtained by the other methods. This suggests that SIRAH may explore a somewhat different region of conformational space compared to the Amber ff19SB and Martini 3 force fields. This is also evident on the energy landscapes (Figure 3), where only in SIRAH there is a large group of conformations, in which the N- and C-termini are very close to each other forming stable interactions. Interestingly, the addition of the disulfide bond in the SIRAH simulations alters the sampled conformations, bringing them much closer to those obtained by the other methods, with fewer conformations comprising contacts between the N- and C- termini. This indicates that the presence of the disulfide bond constrains the conformational flexibility of Aβ42 in the SIRAH model, leading to structures that are more similar to those observed in the Amber ff19SB and Martini 3 simulations. In general, populations of conformations, as characterized by the compactness properties (histograms of R_g_ and D_max_ in Figure 1 and Figure 2, respectively), indicates that the addition of the disulfide bond alters the conformational ensemble towards unimodal distributions, which is especially pronounced in cases of Amber ff19SB and SIRAH, while for the Martini 3 the most important change is the shift towards lower values (or more compact structures).

The analysis of cluster representatives shows that only the all-atom Amber ff19SB force field allows the monomeric Aβ42 to adopt stable secondary structure elements such as β-strands and α-helices (Figure 8). In contrast, such well-defined structures are not present in the representative structures obtained from the coarse-grained SIRAH and Martini simulations. While the SIRAH models do contain structures with some resemblance to β-strands, these secondary structure elements appear distorted and are not recognized by standard secondary structure assignment algorithms such as DSSP [58] or those implemented in PyMol [59] (see Section 3 for more details). The addition of the disulfide bond has the most pronounced effect on the monomeric Aβ42 in Martini, in which none of the cluster-representative structures are almost completely unfolded, contrary to the simulation of the wild-type peptide, where the least abundant ensemble (0.2% of the total population) is almost completely extended.

The presence of the disulfide bond between Cys17 and Cys34 is clearly reflected in the contact map (Figure 9), which shows an increased frequency of contacts in the region surrounding these two residues. Additionally, a small increase in contact frequency is observed between residues 4–8 and 33–37 in Aβ42_disul_, suggesting that the disulfide bond promotes some longer-range interactions between the N- and C-terminal regions of the peptide. In general, the contact map analysis reveals that the introduction of the Cys17-Cys34 disulfide bond results in a notable increase in the number of stabilizing intramolecular interactions in Aβ42 when simulated using the Martini 3 force field.

Another interesting property of the Aβ42 is its secondary structure content (Table 4), which can be compared to the experimental findings (see Section 3 for more details). Interestingly, the addition of a disulfide bond slightly decreased the β-content of Aβ42. Examination of the secondary structure content (Table 4) showed that the Martini simulations failed to predict any semistable secondary structures. A similar effect, but to a smaller extent, was observed for the SIRAH force field. This may be caused by imperfections arising from all-atom reconstruction and secondary structure prediction based mostly on hydrogen bonds, which is, however, the most popular method for determining secondary structures based on PDB files [58].

The incorporation of the disulfide bond decreases the propensity of Aβ42_disul_ to form β-strands by approximately half in the simulations with the Amber ff19SB and SIRAH force fields. However, it does not change the specific regions exhibiting the β-strand propensity. Notably, the disulfide bond reduces the β-strand content in residues 30–35 from around 60–80% to 30–40% in Amber ff19SB and from 40–60% to 20–30% in SIRAH.

Since the monomeric Aβ42 is intrinsically disordered and lacks a stable structure for reference in RMSD calculations, RMSD plots of the initial structure are not presented here. Nevertheless, RMSD for the initial or average structure can be easily calculated with minor modifications of the analysis scripts. On the other hand, we examined how similar the conformations during simulation are to those in fibrils by measuring the RMSD for seven fibril models (U-(2BEG, 2M4J, and 2LMN), S-(5KK3, 2MXU, 2NAO), and LS-(5OQV) shape) deposited in the PDB database (Table 5). Analysis revealed that there is a significant increase in the similarity to fibrils when a disulfide bond is present in the all-atom Amber ff19SB force field. An even larger increase can be observed in the SIRAH force field. On the other hand, results from Martini show that there is a very low similarity of the obtained structure to fibrils, and the addition of a disulfide bond only slightly increases it. This is partially caused by the very large flexibility of the polypeptide chain in the Martini force field, observed in large R_g_ fluctuations, and the ability to temporarily obtain conformations resembling various types of fibrils (Table 6).

## 3. Discussion

In this work, we demonstrated how all-atom and coarse-grained force fields can be utilized to study a model IDP—monomeric Aβ42. To ensure the high reliability of the results, state-of-the-art methods were employed. Specifically, the all-atom Amber ff19SB force field for proteins was used, coupled with the four-point OPC water model [51] and a co-optimized ions model [37] to mimic physiological salt concentration, approximately 0.15 M NaCl [60]. Additionally, the SIRAH coarse-grained force field within the Amber and Gromacs packages and the Martini 3 coarse-grained model and force field within the Gromacs package were employed. In the latter approach, solute-solvent interactions were scaled up and down, and the results were compared to the radius of gyration values to ensure the correct compactness of the simulation structures. The use of a coarse-grained representation significantly accelerated the simulations by approximately 1–2 orders of magnitude, with no significant impact on the results. However, it is worth noting that the use of coarse-grained models can complicate the analysis, as it often requires reconstruction to all-atom models, and some atomistic details are lost due to simplifications of the protein representation. Still, coarse-grained models seem to be a method of choice when it comes to extensive simulations of large IDP complexes [61] and biomolecular condensates [62,63].

Overall, the results obtained indicate that all the methods produced largely unstructured conformations, as expected for the monomeric Aβ42. However, this outcome is not always achieved, especially when older methods are employed [27,64]. Due to the much larger computational cost associated with the all-atom simulations, only three trajectories, each of 2 ms, were executed, which may not provide the highest level of robustness in the results. In contrast, with the coarse-grained SIRAH and Martini methods, simulations were at least 10 times longer, and convergence was reached more quickly due to the reduction in the number of degrees of freedom, resulting in a much lower total computational cost. This approach of employing two entirely different computational methods, such as all-atom and coarse-grained force fields, is recommended for obtaining highly reliable results, especially in the absence of or with limited experimental data.

Earlier computational studies suggest that R_g_ values for Aβ42 are in the range of 0.8 to 1.2 nm [65,66,67], while the experimentally lowest found R_g_ values are about 1 nm, which should correspond to the most compact conformations of the monomers. However, it should be noted that the experimentally observed hydrodynamic radius strongly depends on the conditions and is found to be in the range of 0.8 to 1.5 nm [67], further confirming the molecular flexibility of Aβ42. More recent studies still present very different R_g_ values, ranging from 1.0 [68] to 1.6 nm [69], obtained from SANS experiments. In the context of this evidence, all of the calculated values in this work fit into experimental and theoretical R_g_ ranges, however, all-atom Amber ff19SB and Martini 3 better sample the extended conformations.

The observed average SASA values for the Amber ff19SB force field coupled with the OPC water model of about 38 nm^2^ are significantly larger than the previously obtained SASA values of 32–33 nm^2^ for other Amber force fields coupled with the TIP3P model, while the SASA of about 43 nm^2^ is very close to the results obtained in CHARMM36m, which yielded SASA = 42 nm^2^ [27]. This may also explain why (although Martini shows comparable R_g_ and D_max_ values to Amber ff19SB) the observed SASA values are greater in Martini, as the CHARMM36 force field was used for the backmapping and may contribute to a larger exposure of side chains to solvent.

Analyzing not only average properties but also distributions of their values in the form of histograms and conformational maps provides valuable insights into the structural preferences and dynamic behavior of the system [10,70]. Histograms of such properties as the R_g_ and D_max_ reveal whether the system exhibits a tendency to form semi-stable conformations or explores a wide range of conformations with no strong preferences for specific structures. Energy landscapes, calculated based on the distributions of the end-to-end distance and R_g_, provide information on the relative stability of different conformational states and the energy barriers between them. These analyses are particularly important for IDPs such as the monomeric Aβ42, as they capture the inherent flexibility and conformational diversity of these systems, which may not be adequately represented by average values alone.

Although this manuscript describes most of the standard analyses used for typical studies involving proteins or peptides, more advanced techniques can be employed to gain deeper mechanistic insights, especially when studying complex systems. For example, normal mode analysis (NMA) [71] or essential dynamics [72] can extract the dominant motions from MD trajectories, revealing functionally relevant conformational changes. Markov state models (MSMs) built from MD simulations [73] provide a powerful framework for analyzing the conformational dynamics of proteins, allowing for the identification of metastable states and the kinetics of transitioning between them. This approach is particularly useful for systems exhibiting long timescale processes such as protein folding or oligomerization. Graph-based visualizations based on transition network analysis methods [73,74] can also yield valuable information about the free energy landscape by representing the conformational space as a network, with nodes representing distinct conformational microstates and edges representing the transitions between them. These advanced techniques would be especially beneficial when investigating more complex systems, such as Aβ42 oligomers [75]. The energy landscape in the form of conformational maps of Aβ42 tetramers and higher-order oligomers is highly intricate, featuring multiple association-dissociation events and conformational rearrangements that are challenging to fully capture using standard analyses alone. Integrating NMA, MSMs, and graph-based approaches with conventional methods would provide a more comprehensive understanding of the dynamical behavior and mechanisms underlying the formation and interconversion of these oligomeric species, which are crucial in the pathogenesis of Alzheimer’s disease. However, in the case of the monomeric Aβ42, simple energy landscapes should be sufficient to represent the disordered character of this peptide. This is especially true for the Martini 3 simulations, in which no semi-stable structures are observed. On the other hand, some stable conformations are noticeable in the SIRAH simulations without the disulfide bond; however, they are likely an effect of the Aβ42 overstabilization.

Due to the disordered character of monomeric Aβ42 in water, we cannot compare it to experimental structures in similar conditions. Although the monomeric Aβ42 is disordered in water solution, it can attain relatively stable conformations in apolar environments [76]. As a matter of fact, structures of the monomeric Aβ42 in such conditions can be found in the PDB (see, e.g., entries with the deposition codes 1IYT, 1Z0Q, and 6SZF) [76,77,78]. However, their properties and conformations are significantly different from those in water and cannot be directly compared. For an aqueous environment, only structures of Aβ42 (proto)fibrils are available so far.

To test how accurate the results obtained in this study are compared to the experimental results, we calculated correlations, differences (ΔH), and root-mean-square-errors (RMSE) between chemical shifts predicted for PDB structures from the trajectories and the experimental values [79] (Table 7 and Table 8). This analysis shows that all of the methods agree well with the experiment, with an especially high correlation observed for the Martini simulations without solute-solvent scaling. Not surprisingly, the cysteine mutant with a disulfide bond shows worse agreement with the experimental data than the wild-type Aβ42 peptide; however, the difference is not very large, indicating that this mutation does not significantly alter the properties of the molecule. In general, Amber ff19SB coupled with the OPC water model obtained similar results to the previously tested ff14SB with the TIP3P water model, yet worse than CHARMM36m [27]. It should be noted that these results are significantly better than those obtained using older force fields, such as Amber ff99SB [27] or variants of CHARMM22 [80,81]. According to previous studies, these results could probably be further improved by extending the trajectories even further [82] or using enhanced sampling methods, such as replica exchange molecular dynamics simulations [27].

It should be noted that although chemical shifts provide a measure that can be used for benchmarking force fields for IDPs, such as monomeric amyloid β [80], the correlation between computational and experimental values of the chemical shifts for carbon atoms is almost always higher than for hydrogen and nitrogen atoms, which are encumbered by higher relative errors [83]. While it is understandable for hydrogens, as they are most mobile and therefore often stiffened during simulations to allow the use of a large timestep, or poorly reconstructed from coarse-grained representations, this is not the case for nitrogen atoms, which tend to have too low ΔH values in many all-atom force fields [84], independent of the method used for the chemical shift prediction.

Our results support these previous observations, showing that despite using much different and newer force fields, the chemical shifts for nitrogens are underestimated (Table 8). A direct comparison of the chemical shift values shows that SIRAH, and not Martini, obtained values closest to the experimental ones; however, these values are not as highly correlated with the experimental data as those obtained using the Amber ff19SB force field (Table 7 and Table 8). This observation suggests that while SIRAH may reproduce the magnitude of nitrogen chemical shifts more accurately, the Amber ff19SB and Martini 3 force fields capture the relative variations between residues better, as reflected by the higher correlation coefficients.
ijms-25-06698-t007_Table 7Table 7Correlations between chemical shifts calculated by Shifts 5.6 [85] and those determined experimentally. The correlation coefficients are averaged over trajectories. Standard deviations are given in brackets.Force FieldVariantCαCβNAmber ff19SBwt0.982 (0.007)0.990 (0.004)0.838 (0.036)CC mutant static0.984 (0.003)0.978 (0.004)0.813 (0.029)CC mutant dynamic0.981 (0.006)0.978 (0.001)0.864 (0.038)SIRAHwt0.981 (0.001)0.991 (0.002)0.780 (0.023)CC mutant0.972 (0.004)0.979 (0.003)0.806 (0.037)Martini 3wt0.9920.9950.896CC mutant0.9900.9790.853
ijms-25-06698-t008_Table 8Table 8Difference (ΔH = H_experimental_ − H_computational_) [ppm] and RMSE [ppm] between chemical shifts calculated by Shifts 5.6 [85] and those determined experimentally for the wild-type monomeric Aβ42.Force FieldCαCβNΔHRMSEΔHRMSEΔHRMSEAmber ff19SB0.221.06−0.581.52−2.174.03SIRAH0.030.670.180.96−1.012.60Martini 30.360.750.230.93−6.296.65


The average secondary structure content obtained using Amber ff19SB and SIRAH aligns well with previous experimental findings, which have established the α-helical content to be within the 3–9% range and the β-content to be 12–25% [86]. These values are also similar to those obtained in our previous studies using the CHARMM36m force field [27]. Distributions of the secondary structure content per residue show good agreement with previous theoretical studies (Figure 4). Teplow et al. have observed the α-helical propensity for residues 26–31 and the β-strand propensity for residues 19, 29–33, and 37–39 [87]. Similarly, Strodel et al. have found the α-helical propensity for residues 11–19, 22–26, and the β-strand propensity for residues 16–21 and 30–40 across various force fields [80]. The ranges for the α-helical propensity obtained in this study are very similar to those previously reported for Amber ff14SB using the same sampling methods [27]. However, the β-strand propensity observed here more closely resembles the results from CHARMM36m and ff14SB obtained using replica exchange molecular dynamics (REMD)-enhanced sampling. This suggests that while the ff14SB force field coupled with the TIP3P water model may overstabilize certain conformations, requiring enhanced sampling to accurately capture structural elements and their transitions, the Amber ff19SB force field exhibits proper behavior even with classical MD sampling. The improved performance of ff19SB likely stems from its optimized parameters and the use of the OPC water model, which better represent the balance between protein-protein and protein-water interactions crucial for accurately simulating intrinsically disordered systems such as the monomeric Aβ42. In summary, the secondary structure propensities obtained using Amber ff19SB and SIRAH in this study are consistent with both experimental data and previous computational studies. The results highlight the importance of force field selection and sampling methods in accurately capturing the conformational ensemble of IDPs such as Aβ42.

It should be noted that the presence of distorted secondary structure elements (Figure 8) is not unexpected in coarse-grained simulations because the simplified representation of amino-acid residues, coupled with the use of standard force fields (with no terms for secondary structure corrections), can lead to inaccuracies in local protein structures. This is a commonly known limitation of coarse-graining. Moreover, the procedure used to reconstruct all-atom chains from coarse-grained models is another potential source of structural imperfections. The backmapping from coarse-grained to all-atom representations is a challenging problem and can introduce artifacts [42]. The structural imperfections from backmapping could potentially be alleviated by applying more sophisticated reconstruction procedures, similar to the one employed in Martini 3, which involves a series of short MD simulations with restraints using an all-atom force field such as CHARMM36, rather than relying on the standard backmapping tool such as the one provided by the SIRAH developers. Such restrained MD simulations allow for the reconstructed all-atom model to relax and adopt a more realistic local geometry while still maintaining the overall tertiary structure captured by the coarse-grained simulation. However, this approach may transfer some local tendencies arising from the used all-atom force field, such as the increased SASA values in the case of CHARMM36 and 36m force fields (Table 1).

Point mutations are commonly introduced to proteins and peptides to either decrease or increase their aggregation properties, depending on the study’s objectives. In the case of Aβ42, a series of MD simulations were conducted on the L17C/L34C mutant with a disulfide bond acting as a cross-link between the amino-acid residues that come into contact in the fibrillar form. The designed mutant is known to efficiently form fibrils with both the wild-type and mutated Aβ42, emphasizing some of the fibril-like contacts, including Q15-V36, L17-L34, and F19-I32 [13]. Interestingly, our all-atom and coarse-grained simulations, even without the disulfide bond, exhibit a significant number of conformations forming the latter two contacts, and these conformations were further stabilized when a disulfide bond was incorporated (see Figure 6 and Table 3). We also observed many contacts between the central hydrophobic core (residues 16–22) and the C-terminal region (residues 30–42), indicating the formation of β-hairpins, which were previously shown to be important for the oligomerization of Aβ42 and its variants [88,89]. Surprisingly, despite the higher similarity of the monomeric Aβ42 conformations to the fibrils for the disulfide-bond variant, a decrease in β-content is observed in the Amber ff19SB and SIRAH force fields. The higher propensity of Aβ42 to form fibril-like conformations when a disulfide bond is present seems to be a major reason for the experimentally observed speed-up in aggregation without wild-type fibril seeding in Aβ40 [13], which is consistent with the N* theory that connects the number of fibril-like conformations in the monomeric state with the aggregation rate [90]. This effect, in this case, cannot be simplified to an examination of the impact of β-content on the aggregation rate, which, on the other hand, is probably responsible for the decrease of the aggregation rate when wild-type fibril seeding is used in Aβ40 [13]. These effects also suggest that although the disulfide-bonded L17C/L34C variant can form fibrils with wild-type Aβ, the morphology of these fibrils is not identical, which is typical for disulfide-bond modifications in other amyloids [91,92]. It should be noted that other mutations involving disulfide-bond cross-linking in Aβ may have opposite effects. For example, the introduction of the Cys21-Cys30 disulfide bond prohibited the formation of fibrils; however, it increased the formation of toxic oligomers [93].

The use of the coarse-grained SIRAH model enabled us to obtain approximately a 6-fold decrease in the computational time needed to perform 1,000,000 steps compared to the all-atom ff19SB force field (Table 9). Taking into account a 10 times longer timestep value, this provided a 60-fold speed-up when using a low-end GPU, while the total number of interaction centers decreased by about 13 times. These values are in agreement with those obtained by SIRAH developers for a much larger system [55]. However, it should be noted that in the case of the monomeric Aβ42, the SIRAH simulations on a GPU did not scale at all, even with the use of a much more powerful GPU, while for the all-atom system, a 4-fold speed-up was observed, which most likely was caused by the small size of the studied system. An 80-fold speed-up was observed in the Amber all-atom force field when using a state-of-the-art GPU instead of a 20-core CPU node, whereas this difference was only about 10-fold if SIRAH is used in the Amber package. Interestingly, the SIRAH simulations in Gromacs are significantly faster than the SIRAH simulations in Amber when the CPU is used. This difference may be, at least partially, explained by the use of a simpler cutoff for the Van der Waals interactions in Gromacs (simple cutoff) than in Amber (Particle Mesh Ewald—PME). The dissimilarity disappears when the same GPU is used with PME treatment for all long-range interactions. While this observation may suggest that the CPU performance of Gromacs is superior to that of Amber, further investigations are needed to fully assess the impact of a simpler cutoff approach and to examine various systems on different hardware. It should be noted that the use of the Martini 3 model in the Gromacs package provides about a 15-fold speed-up compared to SIRAH in the Amber package when a similar CPU is used, which most likely comes from a much different coarse-grained model used (especially for water), different cutoff methods, and different ranges. Moreover, the performance of Martini in Gromacs scales well even when a high number of CPU cores is used (e.g., three times speed-up when 64 cores are used instead of 12). It should be noted that the Gromacs package [50] is versatile in the sense that it can be used to perform MD simulations not only with such coarse-grained models as Martini [47] and SIRAH [55], but also within the all-atom model with various force fields, including GROMOS [94], CHARMM [95], and AMBER [34]. Similarly, the Amber package can be used with other than Amber force fields, including CHARMM for proteins and lipids and GAFF [96] for small organic molecules.

In general, all of the tested force fields provided reasonable results in terms of the data that can be experimentally compared, such as the radius of gyration and chemical shifts. SIRAH tends to generate more compact conformations, and fully extended conformations are not observed in it, which may suggest that the method overstabilizes the occurrence of some contacts; however, the obtained properties are still of reasonable quality. On the other hand, Martini tends to generate highly fluctuating conformations within relatively short timescales, and almost no semistable secondary structure elements are observed during the simulations. This effect is probably caused by the fact that for folded proteins, an elastic network coupled with secondary structure parameters is generated in the initial steps of Martini 3, which is necessary to provide the proper secondary structure of proteins [97,98,99,100]. Here, we find that without such bias based on the initial structure, Martini can generate overall conformations of Aβ42 with very high accuracy, as indicated by the comparison of chemical shifts to the experimental data, yet lacking secondary structure detectable by the DSSP and similar methods. Therefore, after the planned optimization of the Martini 3 parameters by the developers, it may be an optimal combination of computational cost and simulation accuracy. However, at this moment, its use is limited if detailed conformational changes are to be observed.

The protocol and description provided here (see Appendix B and Appendix C) can be easily employed for routine simulations of most biomacromolecules, with the flexibility to change the force field, water model, and other components of the system. This adaptability can be managed even by less experienced users. While it is recommended to conduct classical MD simulations on supercomputers, access to which is often limited, the use of GPU computations and coarse-grained representations allows these simulations to be successfully run on modern desktop computers, which are usually equipped with up to 32-core CPUs. This democratizes access to scientific research by eliminating the need for substantial budgets, thereby promoting inclusivity in scientific studies.

## 4. Materials and Methods

Detailed protocols on how the MD simulations and analyses were performed, enabling full reproducibility of the results even by inexperienced users, are provided in Appendix B, along with the exemplary scripts (Appendix A) and a detailed description of their implementation (Appendix C). The representative results discussed in this manuscript are based on three force fields: the all-atom ff19SB force field coupled with the four-point OPC water model, and the coarse-grained SIRAH 2 and Martini 3 force fields coupled with their respective coarse-grained water models. For the all-atom simulations, three independent trajectories, each 2 µs long, were run for each system, providing a total of 6 µs per system. For the coarse-grained simulations, a single 20 µs trajectory was run for Martini, while three 20 µs trajectories were run for SIRAH. Monomeric Aβ42 was simulated in its wild-type form and with L17C and L34C mutations to allow the formation of a disulfide bond, which was previously used as a tool to accelerate Aβ42 aggregation [13]. The disulfide bond was treated as both a static covalent bond and a distance restraint, allowing for breaking and formation during the simulation, as presented in the previous work on dynamic treatment [17].

The simulation protocols for the wild-type and disulfide-bond mutants are described in detail in Appendix B, Options 1–2 for the all-atom simulations and Options 3–4 for the SIRAH coarse-grained simulations using the Amber package, as well as Options 5–6 for the Martini coarse-grained simulations and Option 7 for the SIRAH simulations using the Gromacs package. Option 8 demonstrates the use of CHARMM-GUI to generate initial simulation files.

The coarse-grained models were reconstructed to an all-atom representation using the respective scripts described in the protocol (Appendix B) and its accompanying description (Appendix C). The result of mapping coarse-grained simulation structures back to the all-atom representation is illustrated in Figure 10. Specifically, the final structure of Aβ42_disul_ obtained from the Martini simulation with λ = 1 is shown in the van der Waals representation (panel A), the stick-and-ball representation (panel B), and the ribbon representation (panel C). The all-atom structures obtained from the backmapping calculations can be used not only for visualization purposes but also for analysis and comparison with experimental data [48].

Most of the analysis of the simulation results was performed in AmberTools for simulations run in the Amber package and by using Gromacs tools (e.g., gyrate) as well as Python libraries (including MDTraj [101] and Contact Map Explorer 0.7.0) for Gromacs simulations, together with in-house scripts.

Structural analysis, which requires all-atom conformations (Solvent Available Surface Area (SASA) with Linear Combinations of Pairwise Overlaps (LCPO) method [102], secondary structure with Define Secondary Structure of Proteins (DSSP) method [58] was performed using AmberTools [103] based on the original all-atom or reconstructed all-atom trajectories from the coarse-grained models. Carbon and nitrogen chemical shifts were predicted using the most recent version of the SHIFTS 5.6 tool [85], with default options. Free energy maps were calculated using the following formula:$$\;\Delta F_{n}=-RTln\;\left(I_{n}\;/\;\sum_{a=1}^{N}I_{a}\right)$$
where *R* is the gas constant, *T* is the absolute temperature, $I_{n}$ is the number of structures in *n* bin of 0.02 and 0.1 nm for R_g_ and D_max_, respectively, and $\sum_{a=1}^{N}I_{a}$ is a total number of structures. To determine the most abundant conformations during simulations, a hierarchical agglomerative (bottom-up) clustering method, built in AmberTools’ cpptraj [104], was used.

## 5. Conclusions

In this work, we presented comprehensive protocols for conducting MD simulations on the example of the monomeric form of amyloid-β42 (Aβ42) and its L17C/L34C mutant (Aβ42_disul_). The protocol covers the application of both all-atom and coarse-grained force fields in two widely used simulation packages: Amber and Gromacs. For the all-atom simulations in Amber, we employed the state-of-the-art ff19SB force field coupled with the OPC water model, which has been shown to perform well for both folded and disordered proteins and peptides. We also demonstrated the use of the coarse-grained SIRAH force field in Amber and Gromacs. In Gromacs, we also showcased the application of the broadly-used coarse-grained Martini 3 force field, with corrections for protein-water interactions to better capture the behavior of IDPs, as well as the SIRAH force field. Our protocol not only guides users through the setup and execution of these simulations but also introduces a novel approach for treating disulfide bonds dynamically, allowing them to form and break during the course of the simulation. This dynamic treatment is achieved through the use of distance restraints, providing a more realistic representation of the disulfide bond behavior compared to the conventional static approach. To validate our protocol and demonstrate its utility, we presented new results comparing the conformational dynamics of Aβ42 and Aβ42_disul_ using the various force fields and simulation packages. Our findings clearly illustrate the impact of the disulfide bond on the increased structural similarity between the monomeric form and the fibrillar state of Aβ42, without significantly impacting the secondary structure content. This explains the experimental observation [13] of the increased aggregation rate of the disulfide-bonded Aβ42 mutant. Furthermore, we highlighted the differences in the results obtained from the different methods employed in this study, providing valuable insights into their strengths and limitations for investigating IDPs. In particular, we found that the all-atom Amber ff19SB force field and the coarse-grained Martini 3 force field captured the intrinsically disordered nature of the monomeric Aβ42, while the SIRAH force field overstabilized certain compact conformations. However, all three force fields yield a reasonable agreement with experimental results on such quantities as the radius of gyration and chemical shifts.

By combining the detailed, step-by-step protocols with novel scientific results, this work serves as a valuable resource for researchers interested in studying the dynamics of IDPs and the role of disulfide bonds in their conformational behavior. The protocols can be easily adapted to investigate other IDPs or folded proteins, making them a versatile tool for the broader biomolecular simulation community. Moreover, the insights gained from comparing results obtained with different force fields will help guide force field selection and improvement in future studies of IDPs.

## Figures and Tables

**Figure 1 ijms-25-06698-f001:**
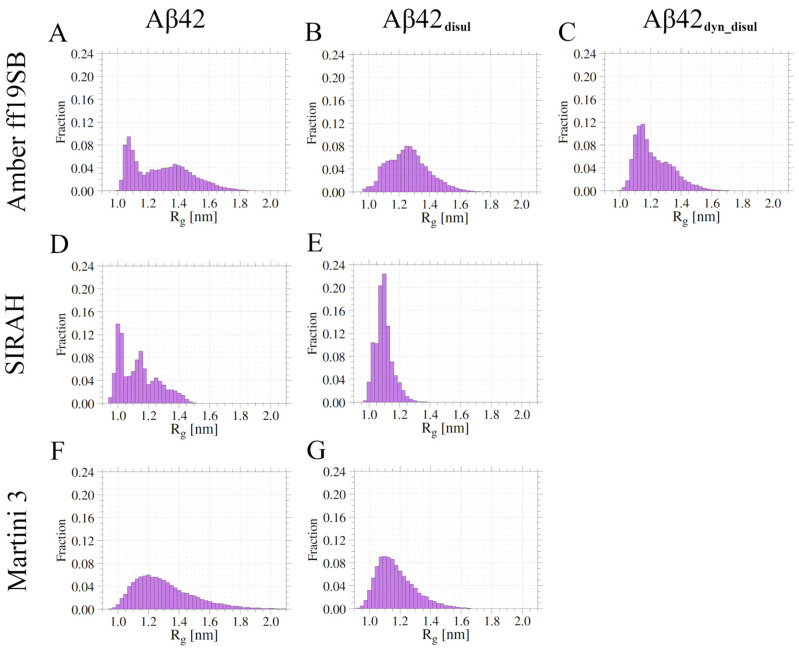
Histograms of R_g_ values obtained from the simulations in Amber ff19SB (panels **A**–**C**), SIRAH (panels **D**,**E**), and Martini 3 (panels **F**,**G**) force fields for wild-type (panels **A**,**D**,**F**) and disulfide-bond mutants (panels **B**,**C**,**E**,**G**).

**Figure 2 ijms-25-06698-f002:**
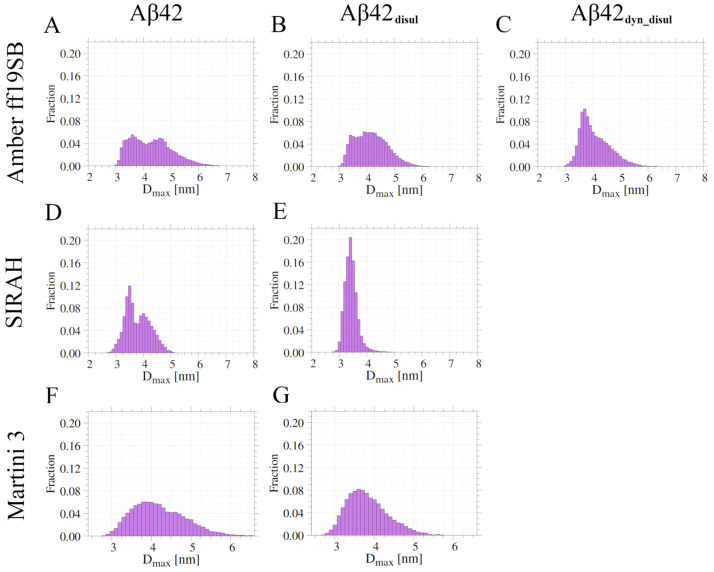
Histograms of D_max_ values obtained from the simulations in Amber ff19SB (panels **A**–**C**), SIRAH (panels **D**,**E**), and Martini 3 (panels **F**,**G**) force fields for wild-type (panels **A**,**D**,**F**) and disulfide-bond mutants (panels **B**,**C**,**E**,**G**).

**Figure 3 ijms-25-06698-f003:**
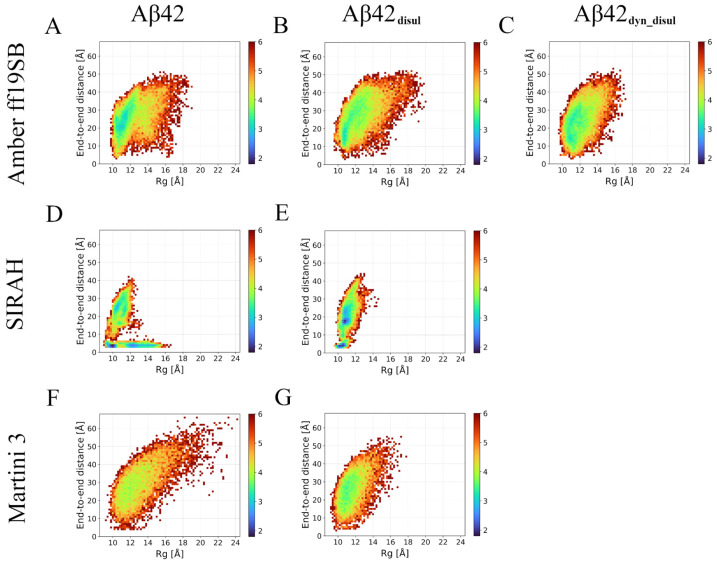
Energy landscapes calculated based on the end-to-end distance and R_g_ values in all trajectories in Amber ff19SB (panels **A**–**C**), SIRAH (panels **D**,**E**), and Martini 3 (panels **F**,**G**) force fields for wild-type (panels **A**,**D**,**F**) and disulfide-bond mutants (panels **B**,**C**,**E**,**G**).

**Figure 4 ijms-25-06698-f004:**
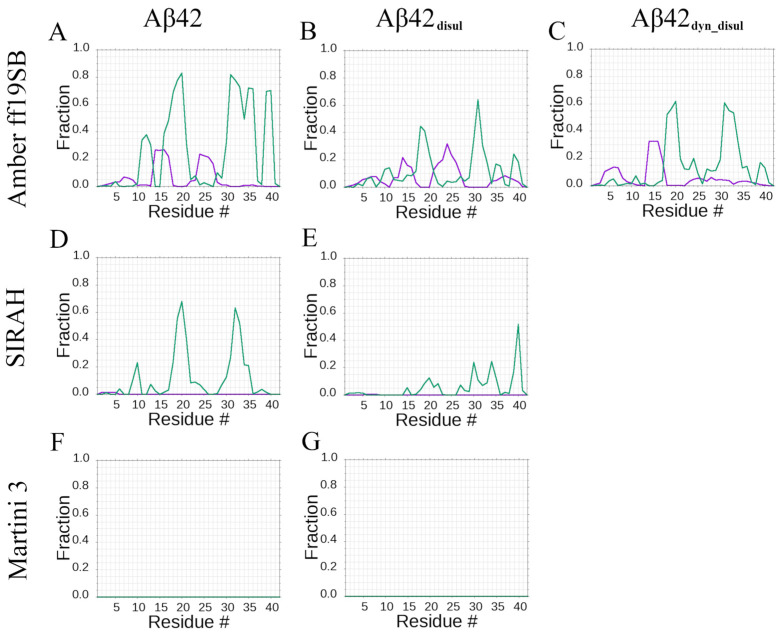
Propensity for amino-acid residues to form β-strands (green) or α-helices (violet), averaged over simulation time and trajectories in Amber ff19SB (panels **A**–**C**), SIRAH (panels **D**,**E**), and Martini 3 (panels **F**,**G**) force fields for wild-type (panels **A**,**D**,**F**) and disulfide-bond mutants (panels **B**,**C**,**E**,**G**).

**Figure 5 ijms-25-06698-f005:**
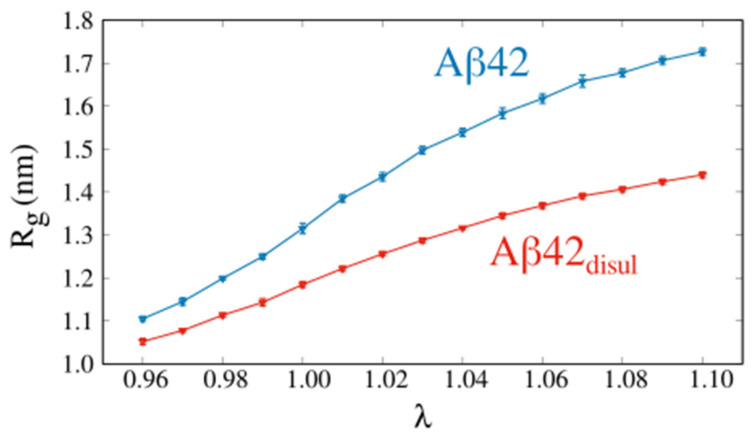
Average radius of gyration, R_g_, as a function of the water-protein interaction rescaling parameter λ obtained from the Martini simulations of Aβ42 (blue) and Aβ42_disul_ (red). The error bars indicate the standard deviation calculated over a single trajectory using block averaging.

**Figure 6 ijms-25-06698-f006:**
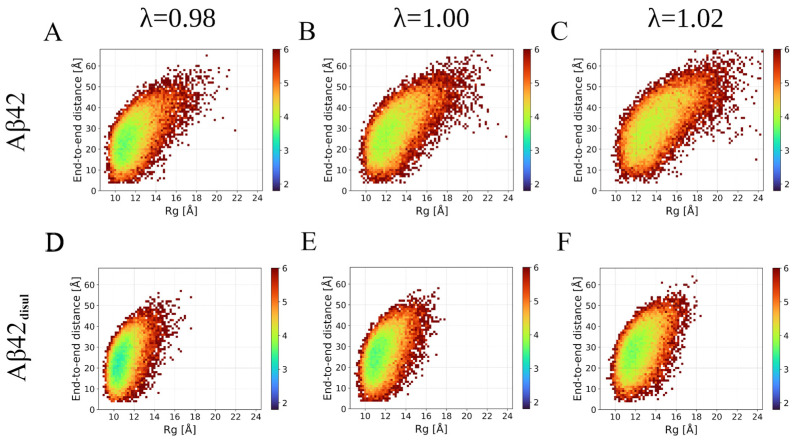
Energy landscapes based on the end-to-end distance and R_g_ distributions obtained from the Martini 3 simulations of Aβ42 (upper panels: **A**–**C**) and Aβ42_disul_ (lower panels **D**,**F**) with the solute-solvent scaling parameter λ set to 0.98 (left panels: **A**,**D**), 1.00 (center panels: **B**,**E**), and 1.02 (right panels: **C**,**F**).

**Figure 7 ijms-25-06698-f007:**
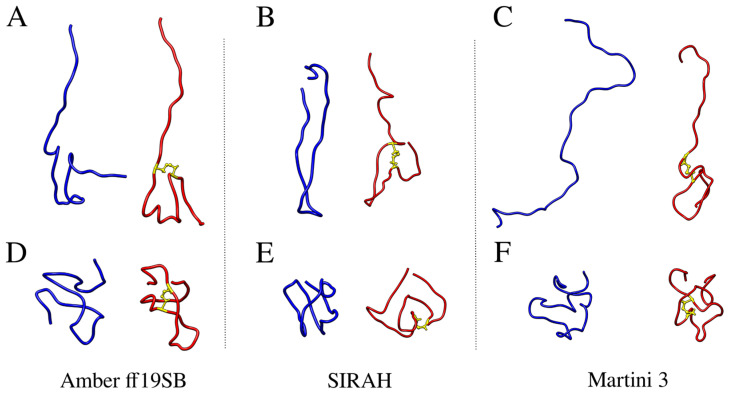
Ribbon representation of the structures from the MD simulations with the all-atom Amber ff19SB (**A**,**D**), coarse-grained SIRAH (**B**,**E**) and coarse-grained Martini 3 (**C**,**F**) force fields with the maximum (**A**–**C**) and minimum (**D**–**F**) R_g_ values shown in blue and red, respectively. Disulfide bonds are marked as yellow sticks.

**Figure 8 ijms-25-06698-f008:**
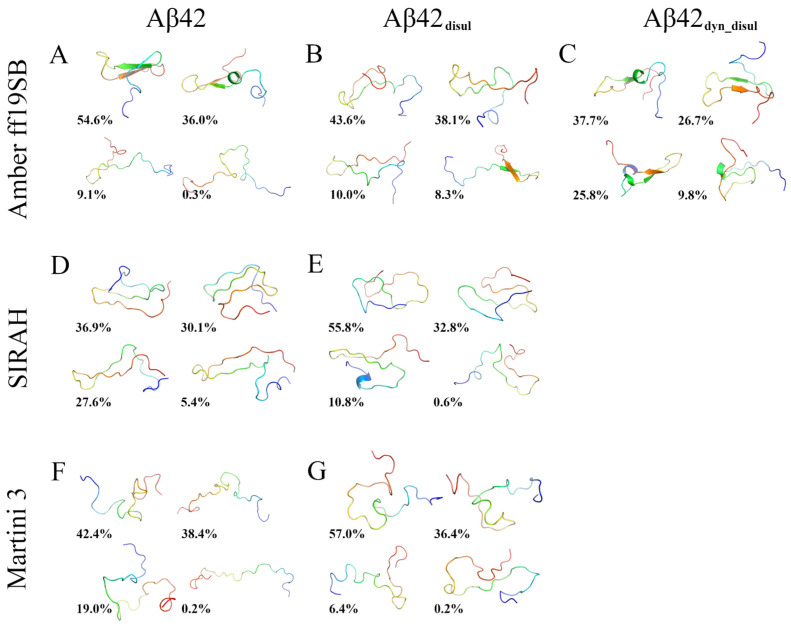
Cartoon representation of the most abundant Aβ42 conformations determined from the clustering analysis in Amber ff19SB (panels **A**–**C**), SIRAH (panels **D**,**E**), and Martini 3 (panels **F**,**G**) force fields for wild-type (panels **A**,**D**,**F**) and disulfide-bond mutants (panels **B**,**C**,**E**,**G**). The population percentages are given below the snapshots. The peptide structure is rainbow colored from blue (N-terminus) to red (C-terminus).

**Figure 9 ijms-25-06698-f009:**
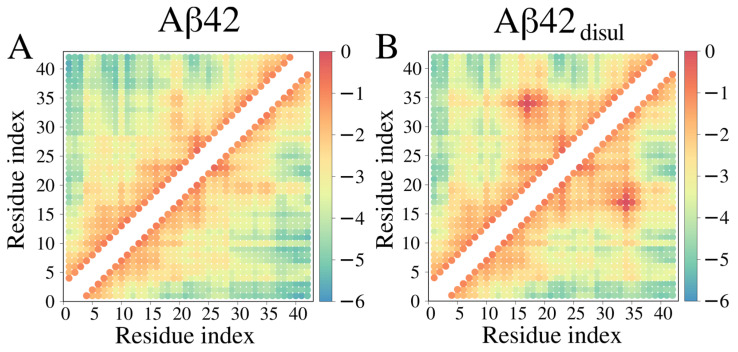
Contact maps obtained from coarse-grained structures of Aβ42 (**A**) and Aβ42_disul_ (**B**) generated in the Martini simulations with λ = 1 (no scaling of solute-solvent interactions). The color scale indicates the natural logarithm of contact frequency. The points in red, yellow, and blue represent frequent, transient, and rare contacts, respectively.

**Figure 10 ijms-25-06698-f010:**
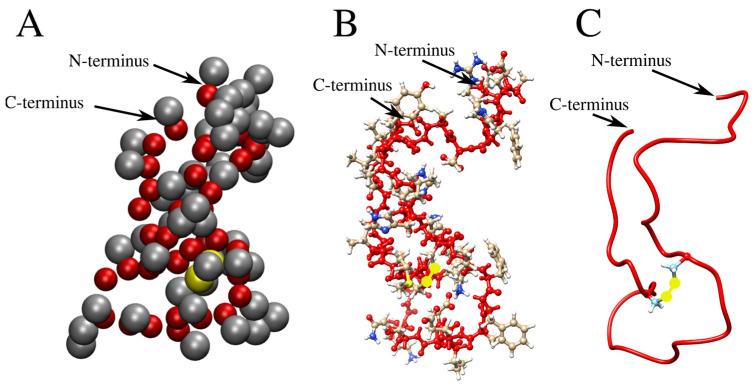
The final structure obtained from the Martini simulation of Aβ42_disul_ is shown in three different representations. (**A**) The coarse-grained structure, where the backbone and sidechain beads are colored in red and gray, respectively. The SC1 beads of Cys17 and Cys34 are highlighted in yellow. (**B**) The all-atom structure, as obtained from the backmapping calculation, is displayed in a stick-and-ball representation. The backbone atoms are shown in red. The sidechain carbon, nitrogen, and oxygen atoms are shown in gray, blue, and red, respectively. The sulfur atoms forming the covalent disulfide bridge are highlighted in yellow. (**C**) The all-atom structure is presented in a ribbon representation: the main chain is shown in red, while sidechains are not displayed, except for Cys17 and Cys34, which are shown in the stick representation with sulfur atoms forming a disulfide bond marked in yellow.

**Table 1 ijms-25-06698-t001:** Average R_g_, D_max_, and SASA with SD (for the Amber and SIRAH force fields, the averages and SDs are calculated over three trajectories, while for Martini it is calculated over a single trajectory with block averaging).

	Amber ff19SB Aβ42	Amber ff19SB Aβ42_disul_	Amber ff19SB Aβ42_dyn-disul_	SIRAH Aβ42	SIRAH Aβ42_disul_	Martini Aβ42 with λ = 1	Martini Aβ42_disul_ with λ = 1
R_g_ (nm)	1.29 ± 0.14	1.27 ± 0.01	1.22 ± 0.07	1.13 ± 0.11	1.10 ± 0.04	1.32 ± 0.01	1.19 ± 0.01
D_max_ (nm)	4.22 ± 0.42	4.16 ± 0.06	4.09 ± 0.21	3.84 ± 0.32	3.48 ± 0.09	4.20 ± 0.04	3.83 ± 0.03
SASA (nm^2^)	37.67 ± 2.78	38.57 ± 0.92	37.73 ± 1.89	36.47 ± 2.63	38.08 ± 0.74	42.90 ± 3.45	39.54 ± 3.87

**Table 2 ijms-25-06698-t002:** Average R_g_, D_max_, and end-to-end distance with SD calculated over a single trajectory in Martini 3 with the solute-solvent scaling parameter λ set to 0.98, 1.00, and 1.02.

	Martini Aβ42 with λ = 0.98	Martini Aβ42 with λ = 1.00	Martini Aβ42 with λ = 1.02	Martini Aβ42_disul_ with λ = 0.98	Martini Aβ42_disul_ with λ = 1.00	Martini Aβ42_disul_ with λ = 1.02
R_g_ (nm)	1.20 ± 0.16	1.32 ± 0.19	1.44 ± 0.22	1.11 ± 0.10	1.18 ± 0.12	1.26 ± 0.13
D_max_ (nm)	3.85 ± 0.59	4.20 ± 0.67	4.52 ± 0.71	3.59 ± 0.47	3.83 ± 0.52	4.06 ± 0.55
End-to-end distance (nm)	2.65 ± 0.94	3.00 ± 1.05	3.32 ± 1.11	2.47 ± 0.83	2.70 ± 0.90	2.93 ± 0.94

**Table 3 ijms-25-06698-t003:** Frequency (in % with SD) of contacts between selected amino-acid residues averaged over trajectories. A contact between two amino acid residues is counted if the Cβ atoms in the all-atom representation or centers of interactions in coarse-grained representations are within a 0.7 nm range.

	Amber ff19SB Aβ42	Amber ff19SB Aβ42_disul_	Amber ff19SB Aβ42_dyn-disul_	SIRAH Aβ42	SIRAH Aβ42_disul_	Martini Aβ42	Martini Aβ42_disul_
Q15-V36	12.54 ± 6.40	9.09 ± 9.94	1.55 ± 1.67	3.45 ± 2.65	0.00 ± 0.00	1.63 ± 0.35	7.11 ± 0.63
L17-L34 (C17-C34)	59.10 ± 29.75	100.00 ± 0.00	100.00 ± 0.00	43.35 ± 7.25	100.00 ± 0.00	5.39 ± 0.75	100.00 ± 0.00
F19-I32	82.34 ± 10.27	58.20 ± 41.84	70.05 ± 36.89	25.72 ± 22.65	26.88 ± 36.38	12.02 ± 0.92	24.72 ± 1.09

**Table 4 ijms-25-06698-t004:** Average secondary content and standard deviations (in brackets) for each of the trajectories.

Force Field	Variant	α-Helices	β-Strand	Other
Amber ff19SB	wt	5.4 (7.3)	23.9 (11.3)	70.7 (7.0)
	CC mutant static	7.7 (3.4)	9.8 (4.6)	82.5 (7.9)
	CC mutant dynamic	5.6 (2.0)	13.1 (8.6)	81.2 (9.3)
SIRAH	wt	0.1 (0.2)	11.4 (3.5)	88.4 (3.7)
	CC mutant	0.0 (0.1)	5.4 (2.8)	94.5 (2.8)
Martini 3	wt	0.8 (2.5)	0.5 (1.6)	98.7 (3.0)
	CC mutant	0.8 (2.5)	0.6 (1.8)	98.6 (3.1)

**Table 5 ijms-25-06698-t005:** Percentage of snapshots with CαRMSD < 6 Å to selected fibril models and standard deviations (in brackets) averaged over trajectories calculated for residues 17–40.

Force Field	Variant	2BEG	2LMN	2M4J	2MXU	2NAO	5KK3	5OQV
Amber ff19SB	wt	0.0 (0.0)	6.3 (10.2)	43.2 (17.6)	2.6 (2.2)	5.9 (4.1)	6.4 (4.9)	2.2 (1.5)
	CC mutant static	0.1 (0.1)	2.6 (3.7)	20.2 (13.8)	34.4 (12.7)	53.7 (20.2)	53.0 (21.2)	31.9 (12.0)
	CC mutant dynamic	1.6 (2.7)	2.3 (2.6)	43.8 (28.7)	12.3 (11.7)	29.4 (13.9)	21.1 (4.5)	7.7 (6.7)
SIRAH	wt	0.0 (0.0)	0.0 (0.0)	10.8 (10.3)	3.6 (6.2)	29.7 (46.5)	8.8 (14.4)	0.3 (0.6)
	CC mutant	0.0 (0.0)	0.0 (0.0)	64.7 (56.0)	26.1 (45.3)	93.1 (5.1)	75.2 (16.4)	20.6 (30.6)
Martini 3	wt	2.5	2.8	13.2	8.5	11.4	8.7	5.5
	CC mutant	1.3	2.3	25.8	7.5	18.1	13.5	5.0

**Table 6 ijms-25-06698-t006:** Lowest RMSD values for selected fibril models calculated for residues 17–40.

Force Field	Variant	2BEG	2LMN	2M4J	2MXU	2NAO	5KK3	5OQV
Amber ff19SB	wt	6.39	5.20	4.24	4.23	4.07	4.06	4.66
	CC mutant static	5.74	4.51	3.46	3.78	3.31	3.28	4.03
	CC mutant dynamic	4.87	4.89	3.68	3.69	3.61	3.60	4.04
SIRAH	wt	6.27	6.22	5.46	5.55	4.69	5.34	5.90
	CC mutant	7.02	6.94	4.48	4.18	3.52	3.86	4.84
Martini 3	wt	3.95	4.40	3.38	3.31	3.82	3.80	3.62
	CC mutant	5.12	4.68	3.52	3.96	3.18	3.34	3.97

**Table 9 ijms-25-06698-t009:** Computational times for various all-atom and coarse-grained approaches for the monomeric Aβ42 with and without disulfide bond treatments are shown with multiple configurations to demonstrate that these simulations can be run on a modern PC or a computer cluster.

Software	Force Field	Disulfide Bond	Number of Interaction Centers:		Timestep, Cutoff Values (Method to Calculate van der Waals/Coulomb Interactions)	Real Time for 1,000,000 Steps	Hardware	
			Peptide	Total			CPU/GPU Model	Core No. and Frequency
Amber 22	All-atom Amber ff19SB	none	627	60,718	2 fs 0.9 nm (PME/PME)	580 min 0 s	2*Intel(R) Xeon(R) E5-2670 v2	20 cores @ 2500 MHz
						8 min 50 s	NVIDIA A100-SXM4-40GB	6912 cores @ 1410 MHz
						38 min 21 s	GeForce GTX 1660 SUPER	3584 cores @ 1800 MHz
		static	609	62,396	2 fs 0.9 nm (PME/PME)	6 min 50 s	NVIDIA GeForce RTX 4090	16,384 cores @ 2520 MHz
						6 min	NVIDIA A100-SXM4-40GB	6912 cores @ 1410 MHz
						40 min 13 s	GeForce GTX 1660 SUPER	3584 cores @ 1800 MHz
		dynamic	611	62,378	2 fs 0.9 nm (PME/PME)	10 min	NVIDIA A100-SXM4-40GB	6912 cores @ 1410 MHz
						40 min 20 s	GeForce GTX 1660 SUPER	3584 cores @ 1800 MHz
	Coarse-grained SIRAH	none	195	4774	20 fs 1.2 nm (PME/PME)	50 min 20 s	2*Intel(R) Xeon(R) E5-2670 v2	20 cores @ 2500 MHz
						4 min 35 s	NVIDIA GeForce RTX 4090	16,384 cores @ 2520 MHz
						3 min 40 s	NVIDIA A100-SXM4-40GB	6912 cores @ 1410 MHz
						9 min 45 s	GeForce GTX 1660 SUPER	3584 cores @ 1800 MHz
		static	195	4926	20 fs 1.2 nm (PME/PME)	3 min 40 s	NVIDIA A100-SXM4-40GB	6912 cores @ 1410 MHz
						7 min 10 s	GeForce GTX 1660 SUPER	3584 cores @ 1800 MHz
Gromacs 2023.5	Coarse-grained MARTINI 3	none	96	6722	20 fs 1.1 nm (cutoff/reaction-field)	1 min 07 s	AMD EPYC 7763	64 cores @ 2450 MHz
						4 min 20 s	Intel(R) Core(TM) i7-5820K	6 cores @ 3300 MHz
						1 min 22 s	GeForce GTX 1660 SUPER	3584 cores @ 1800 MHz
		static	96	6770	20 fs 1.1 nm (cutoff/reaction-field)	5 min 5 s	Intel(R) Core(TM) i7-5820K	6 cores @ 3300 MHz
						1 min 19 s	GeForce GTX 1660 SUPER	3584 cores @ 1800 MHz
	SIRAH	none	195	4644	20 fs 1.2 nm (cutoff/PME)	1 min 43 s	AMD EPYC 7763	64 cores @ 2450 MHz
						4 min 37 s	AMD Ryzen 7 5700G	8 cores @ 3800 MHz
						1 min 59 s	GeForce GTX 1660 SUPER	3584 cores @ 1800 MHz
		static	195	4644	20 fs 1.2 nm (PME/PME)	2 min 53 s	AMD EPYC 7763	64 cores @ 2450 MHz
						7 min 38 s	AMD Ryzen 7 5700G	8 cores @ 3800 MHz
						9 min 38 s	GeForce GTX 1660 SUPER	3584 cores @ 1800 MHz

## Data Availability

All of the scripts and initial files necessary to perform the simulations are included in the Appendix A. Additional data is available upon request from the corresponding author.

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
