# Peer review of "A Practical Guide to All-Atom and Coarse-Grained Molecular Dynamics Simulations Using Amber and Gromacs: A Case Study of Disulfide-Bond Impact on the Intrinsically Disordered Amyloid Beta"

_ijms, 2024, doi:10.3390/ijms25126698_

Round 1

Reviewer 1 Report

Comments and Suggestions for Authors

This work presents a comprehensive protocol for conducting molecular dynamics simulations of amyloid beta (Aβ42), an intrinsically disordered protein (IDP), with and without disulfide bonds. The protocol utilized both all-atom (Amber) and coarse-grained (Martini 3 and Sirah) approaches with the Amber and Gromacs packages. The study reveals the interactions between the monomer Aβ42 and its mutants with the solvent, as well as the effects of disulfide bonds in their structures. The manuscript is well structured and the results are presented in a clear way. Thus, I recommend its publication with the following minor issues.

1. In lines 160-162 on page 5, what is “the maximum dimension (Dmax)”? Is it the same as “the maximum distance between any of the solute atoms and compare it with the box”?

2. How about the comparison of the kinetic results from different simulations using, e.g., Markov state models?

Reviewer 2 Report

Comments and Suggestions for Authors

The manuscript authored by Smardz et al. demonstrates the effect of disulfide bond in the conformation of Ab42 monomer and its mutants based on all-atom and coarse-grained molecular dynamics simulations. Though the manuscript might be of interest to the community, the manuscript has to be significantly revised according to following major concerns:

1. The title should be changed. The authors employed all-atom and coarse-grained molecular dynamics simulations with different force fields to study the conformations of Ab42 monomer and its mutant under the effect of disulfide bond. I suspect that the authors did not develop any force field or coarse-grained model, but they just employed the currently available force fields and coarse-grained models for molecular dynamics simulations. The term “Multi-Scale Molecular Dynamics” is inappropriate, because the authors did not develop any multiscale modeling-based simulation techniques or did not couple all-atom model to coarse-grained model in order to develop a multiscale model.

2. Though the authors showed the trajectory of maximal imaged distance, they should elaborate the numerical simulation results to properly show the information of conformations for Ab42 monomer, its mutant, and Ab42 monomer with disulfide bond. For example, the authors should show the map based on radius of gyration and end-to-end distance with showing the conformations corresponding to the high population. For example, see Fig. 1 in the paper [Levine et al., PNAS 112, p.2758 (2015)], where the conformation map based on radius gyration and end-to-end distance is shown. Or, the authors can show the conformation map based on two relevant order parameters (corresponding to two beta-strands formed in the aggregation form of Ab42 monomer). For example, see Fig. 3 in the paper [Choi et al., PCCP 23, p.22532 (2021)]. Overall, the authors should show the conformation map based on relevant measures with showing the specific conformation corresponding to highly populated structure.

3. As commented above, the author should show the highly populated conformations of Ab42 monomer rather than showing conformations corresponding to the minimum and maximum values of radius of gyration shown in Fig. 6. Instead, the authors can show the highly populated conformations of Ab42 monomer based on the clustering method with using MD trajectories.

4. When the authors show the time-dependent values of measured quantities (e.g. radius of gyration), they should show not only the time-dependent values but also show the histogram, which allows for further insight into the conformations of Ab42. For instance, if there are two peaks in the histogram for radius of gyration, then there might be two main conformations corresponding to these two peaks of radius of gyration. Overall, the authors are recommended to show the histogram along with time-dependent values.

5. Though the authors made a comparison between all-atom and coarse-grained molecular dynamics simulations in the aspect of computational costs, there is no discussion with comparison between all-atom and coarse-grained molecular dynamics simulations in the viewpoint of the highly populated conformations of Ab42 monomer (with/without disulfide bond) and its mutant. Does the coarse-grained molecular dynamics simulation capture the features of highly populated conformations of Ab42 monomer obtained from all-atom molecular dynamics simulation? The authors should properly discuss how the coarse-grained molecular dynamics simulation is working for prediction of conformations for Ab42 monomer.

Comments on the Quality of English Language

The quality of language seems ok.

Reviewer 3 Report

Comments and Suggestions for Authors

This manuscript reports all-atom and coarse grain molecular dynamics simulations of the amyloid β42 using different force fields in Amber and Gromacs. To a large extend the manuscript reads like a technical note and not like a scientific publication. I cannot find any new scientific insight or theoretical development in the manuscript. Therefore, I find the manuscript unsuitable in its present form for publication in Int. J. Mol. Sci.

On the other hand, the manuscript offers practical guidance for molecular dynamics simulations that might be useful for a specialized readership. Thus, publication in a more specialized journal seems appropriate.

Minor points:

· What is the difference between “fluxional” and “intrinsically flexible” molecules?

· Is it not more appropriate to simulate the disulfide bond breaking with a QM/MM approach?

· Why are the nitrogen chemical shift correlation so much worse than the carbon ones?

Reviewer 4 Report

Comments and Suggestions for Authors

I have read the manuscript and I have appreciated the idea to compare the performance of three different simulation protocols in investigating the conformational dynamics of the famous disordered Ab1-42 peptide (which forms fibrils that are involved in Alzheimer disease), and its L17C/L34C mutant (i.e., in which a disulfide bridge has been inserted). On both proteins, the authors have done all-atom MD simulations in the Amber package, using the ff19SB force-field coupled to the OPC water model, and coarse-grained MD simulations in the Gromacs package, using SIRAH 2 and Martini 3 force-fields, coupled to the respective coarse-grained water models. The analysis is interesting because it provides a clear example of the care that must be used in performing MD simulations on intrinsically disordered peptides and proteins, and in trying to define useful constraints to ensure the reliability of the results. The authors have also written two Appendices that provide a detailed step-by-step description-explanation of the different protocols devised to run MD simulations.

I think that the manuscript deserves to be published as it stands.

Author Response

We thank the Reviewer for the positive assessment of our work.

Round 2

Reviewer 2 Report

Comments and Suggestions for Authors

As the authors addressed all the issues raised by the reviewer, the revised manuscript seems to be acceptable for publication.

Author Response

We thank the Reviewer for the positive feedback.

Reviewer 3 Report

Comments and Suggestions for Authors

According to the authors of the manuscript the unusual form of it has been communicated to the IJMS editorial office and has been accepted. Therefore, I withdraw my criticism of the manuscript form.

The minor technical points I raised in my original review have been almost all most satisfyingly resolved. The only exception is the reference to Car-Parinello molecular dynamics in QM/MM. It is nowadays well established that CPMD is not suitable for most applications and should be substituted by Born-Oppenheimer MD. Note that this is now also the method of choice in CP2K. Therefore, I strongly recommend substituting this reference by a more state-of-the-art reference to BOMD as used in QM/MM methods (e.g. Molecules 24, 1653, 2019 or similar). A good overview is given in the book “Multiscale Dynamics Simulations: Nano- and Nano-bio Systems in Complex Environments, Editors: D.R. Salahub, D. Wei, RSC Theoretical and Computational Chemistry Series, London (2021)”. With this change the manuscript is now suitable for publication.

Author Response

We thank the Reviewer for the comments. We replaced the CPMD reference in the revised manuscript with the modified CP variant, which is also used in CP2K. We also added information about the BOMD together with the references for relevant papers, both those mentioned by the Reviewer and alternative methods (e.g., Quantum ESPRESSO). The changed sentence is marked in orange in the revised manuscript and pasted below:

"It should be noted that this approach is of significantly lower resolution than ab initio MD[18, 19], such as Born-Oppenheimer or second-generation Car–Parrinello MD[20] implemented in software packages like CP2K[21] or Quantum Espresso[22], quantum mechanics/molecular mechanics (QM/MM), which can be run in Amber and Gromacs combined with the interface of CP2K or other tools, like Quick[23], and reactive force field approaches, such as ReaxFF [24–26]."